# The Formulation of the N-Acetylglucosamine as Nanoparticles Increases Its Anti-Inflammatory Activities: An In Vitro Study

**DOI:** 10.3390/bioengineering10030343

**Published:** 2023-03-09

**Authors:** Alessia Mariano, Irene Bigioni, Sergio Ammendola, Anna Scotto d’Abusco

**Affiliations:** 1Department of Biochemical Sciences, Sapienza University of Rome, 00185 Roma, Italy; 2Ambiotec di Sergio Ammendola, 04012 Cisterna di Latina (LT), Italy

**Keywords:** N-acetylglucosamine, inflammation, nanoparticles, osteoarthritis, chondrocytes

## Abstract

Nanomedicine can represent a new strategy to treat several types of diseases such as those with inflammatory aetiology. Through this strategy, it is possible to obtain nanoparticles with controlled shape, size, and eventually surface charge. Moreover, the use of molecules in nanoform may allow more effective delivery into the diseased cells and tissues, reducing toxicity and side effects of the used compounds. The aim of the present manuscript was the evaluation of the effects of N-acetylglucosamine in nanoform (GlcNAc NP) in an in vitro model of osteoarthritis (OA). Human primary chondrocytes were treated with Tumor Necrosis Factor (TNF)-α to simulate a low-grade inflammation and then treated with both GlcNAc and GlcNAc NP, in order to find the lowest concentrations able to counteract the inflammatory state of the cells and ensure a chondroprotective action. The findings showed that GlcNAc NP was able to decrease the pro-inflammatory mediators, IL-6 and IL-8, which are among the main effectors of inflammation; moreover, the nanoparticles downregulated the production of metalloprotease enzymes. GlcNAc NP was effective at a very low concentration compared to GlcNAc in its native form. Furthermore, GlcNAc NP stimulated an increase in collagen type II synthesis. In conclusion, the GlcNAc in nanoform showed better performance than GlcNAc, at concentrations lower than those reached in the joints after oral administration to patients of 1.5 g/die of glucosamine.

## 1. Introduction

Many diseases share inflammation as a common feature, which is the immune response able to ensure survival and tissue homeostasis. However, the inflammatory processes can become detrimental both in acute and chronic phases, contributing to the onset of diseases [1]. Pulmonary viral infections, mainly if associated with cytokine storm, represent an example of acute inflammation [2], whereas joint disease such as osteoarthritis represents chronic inflammation [3], meaning that inflammatory processes can be stimulated both by external pathogens and by endogenous response. Nanotechnological approaches could represent an innovative strategy to treat diseases with inflammatory aetiology. Osteoarthritis (OA) is a connective tissue disease regarding joints and shows an inflammatory aetiology that leads to the degradation of cartilage [4]. OA is mainly treated with non-steroidal anti-inflammatory drugs and with compounds such as glucosamine (GlcN) and N-acetiylglucosamine (GlcNAc), which are considered symptomatic slow-acting drugs for osteoarthritis [5,6].

Nevertheless, despite a variety of promising disease-modifying OA compounds, the availability of drugs to modify cartilage degeneration remains challenging. Bioavailability and drug delivery are the most important causes of failure. In addition, epigenetic changes plus individual polymorphisms increase the variability of response to treatments [7]. This made it necessary to increase the drug dosage to ensure the pharmacological response. However, strategies to increase the bioavailability of registered drugs remain an active field of study. To date, intra-articular injection of drugs is the most used method to overcome these problems [8]. Alternatively, biomaterials have also been studied to improve the bioavailability of GlcN [9]. To this aim, new formulations, such as nanomaterials using hydrogels or hybrid exosomes, are under study [10,11,12]. Reformulation of registered drugs such as analgesics, glucocorticoids, and hyaluronic acid, are the most used to improve OA therapies. Along with these therapies, the use of Glucosamine (GlcN) and its derivatives has become widespread in the last few decades, especially as food supplements added to a composition of nutraceuticals [13]. Clinical trials confirmed that glucosamine sulphate, at high doses, is the most active molecule to treat OA pain [14,15]. On the other hand, any current reformulation of GlcN was poorly able to increase drug delivery or efficacy [16]. The effective dosage is about 2 g/day (or 100 mM), but GlcN was found to induce autophagy both in vitro and in vivo chondrocytes [17]. To reduce the amount, GlcN is used along with chondroitin sulphate [18], or with alendronate [19]. Among GlcN derivatives, GlcNAc, a natural GlcN metabolite, is used to treat OA. Studies on the amino-sugar transport in chondrocytes showed that GlcN enters mainly via GLUT2, whereas GlcNAc does not enter these cells, affecting their functions probably through the binding to GLUTs [20]. Thus, GlcN uptake inhibits the facilitated transport of glucose, affecting the homeostasis of chondrocytes, whereas GlcNAc stimulates glucose transport [20]. The use of GlcNAc to treat OA has been studied on cartilage metabolism in healthy people using 500 mg/day and 1000 mg/day dosages, obtaining good effectiveness [21]. Moreover, encouraging results were obtained also in a rat OA model where, after oral administration of GlcNAc for 28 days (1000 mg/kg/day), an improvement of damaged tissue was observed [22]. To improve delivery and distribution, GlcNAc has been intra-articular injected in a rabbit model of OA, using cell-free porous poly lactic-co-glycolic acid (PLGA) graft implants. The results showed that the acetyl-aminosugar promoted the repair of full-thickness articular cartilage defects in the animal joints and that the effect is prolonged or increased when a combination of compounds was used [23]. Moreover, GlcNAc has been shown to exert anti-inflammatory activity and the ability, albeit of limited potency, in stimulating the synthesis of certain factors whose degradation is associated with OA. Finally, prolonged administration of this molecule at high dosages, up to 4 g/day, may induce side effects, especially in the elderly [24]. The traditional method of industrial production of GlcNAc is based on the acid hydrolysis of crab and shrimp shells, but recently a bioproduction of this molecule has been achieved using different microorganisms [25]. These types of processes yield a micrometric powder of particles having a medium size between 0.025 mm and 1 mm. The present paper aims to describe a method to comminute the GlcNAc granules in nanoparticles (GlcNAc NP) by planetary ball mill, in order to improve the anti-inflammatory effects and decrease the administered amount. The process yields nanoparticles with an average size of 200 nm and a stable polydispersity index, which showed the ability to down-regulate the pro-inflammatory mediator production alongside the collagen II stimulation. The findings showed that the GlcNAc NPs are effective at very low concentrations compared to their native form. To our knowledge, this is the first description of the preparation of GlcNAc in nanoform by planetary ball mill and the first time that these nanoparticles are used to study the anti-inflammatory and anabolic activity in cartilage degradation in vitro model.

## 2. Materials and Methods

### 2.1. Materials

N-acetylglucosamine raw powder and polyvinylpyrrolidone (PVP, E1201 additive), both certified for human use, were obtained from ACEF SpA (Fiorenzuola D’Arpa, PC, Italy). Dulbecco’s modified eagle medium (DMEM) High Glucose and Phosphate Buffer Saline (PBS) were purchased from HyClone (Logan, UT, USA); L-glutamine, penicillin/streptomycin, gentamicin, Fetal Bovine Serum (FBS) and Triton 100 were all obtained from Sigma Aldrich (St. Louis, MO, USA). MTS (3-(4,5-dimethylthiazol-2-yl)-5-(3-carboxymethoxyphenyl)-2-(4-sulfophenyl)-2H-tetrazolium)-based colourimetric assay and reverse transcription enzyme, Improm II, were purchased from Promega (Promega Corporation, Madison, WI, USA). Blood/Tissues Total RNA extraction kit was purchased from Fisher Molecular Biology (Trevose, PA, USA), SensimixPlus SYBR Master mix was purchased from Bioline (London, UK), and primers were synthesized by Bio-Fab research (Rome, Italy). Antibody anti-collagen type II was obtained from Santa Cruz Biotechnology (Dallas, TX, USA). Alexa Fluor 595 donkey anti-mouse red secondary antibody and DAPI, the dye used to stain the nuclei, were both purchased from Invitrogen (Thermo Fisher Scientific, Waltham, MA, USA). IL-6 and IL-8 Enzyme-Linked Immunosorbent assay (ELISA) kits were purchased from Fine Test (Fine Biotech Co., Ltd., Wuhan, China).

### 2.2. Preparation of the NanoGlcNAc

The N-acetylglucosamine raw powder was analysed to determine its granulometry by certified sieves (Filtra Vibraciòn, Barcelona, ES, Spain). To set up the synthesis, many tests were necessary, for which the powder/balls ratio, the size of the balls, and the time and the speed of milling were varied. The best conditions were obtained by filtrating the dry powder on a 45-mesh sieve and after mixing (20:1) 20 g of powder with PVP certified for human use (E1201 additive). The mixture was dissolved 1:3 (*v*/*v*) in ultrapure water and milled in a zirconium jar by a planetary ball mill. One volume of zirconium balls, with a diameter of 2 mm, was added to the suspension and the process was performed for 7 h at room temperature. The wet mixture was recovered and dried under a vacuum oven and stored at room temperature for further analyses and experiments.

### 2.3. Physical Characterization of Nanoparticles

The dried nanoparticles were dispersed in ultrapure water at a concentration of 5 mg/mL and filtered on a 0.22 μm PTFE membrane. The dispersion was stirred until complete dissolution and ultrasonicated for 5 min in a water bath. Samples from three independent preparations were analysed by a Dynamic Light Scattering (DLS, Zetasizer Ultra Malvern-Panalytical). The size was analysed at a fixed scattering angle of 633 nm, according to the Siegert relation to measuring the hydrodynamic size [26]. Each sample was analysed by using a disposable capillary cell ZSU1002, and the scattered radiation was measured using the Extended Size Range modality. The samples were left for 120 s at 25 °C and then analysed by setting a dynamic viscosity at 0.8872 cP, a dispersant refractive index of 1.33, and fixed coaxial at the centre of capillary cell 4.64.

Moreover, the concentration of nanoparticles of different sizes was analysed in the colloidal dispersion by performing a Multi-Angle Dynamic Light Scattering (MADLS). The analysis was performed at three scattering angles: Backscatter 173° (NIBS), Side scatter 90° (DLS) and Forward scatter 17°. The scattered radiations were measured using disposable polystyrene cuvettes DTS0012. In all experiments, the measurement position was automatic for MADLS, whereas the correlation time was an adaptive correlation.

Hydrodynamic diameter or Z-average was estimated according to ISO13321 by fitting Cumulants and Polydispersity Index. The size distribution was estimated with a different algorithm ZS XPLORER according to the constructor’s instruction.

### 2.4. Human Primary Cell Isolation and Culture

Human primary chondrocytes (HPCs) were isolated from OA patients that underwent surgical treatment for knee or hip replacement, and full ethical consent was obtained from all donors. The study was approved by the Research Ethics Committee, Sapienza University of Rome (#290/07, 29 March 2007), and ASL Lazio 2 (#005605/2019, 3 March 2019).

HPCs were isolated as previously described [27]; briefly, femoral and tibial condyles and femoral heads from articular cartilages were aseptically dissected from patients, selecting areas of macroscopically normal cartilage. Cartilage was minced and digested for 30 min with 1 mg/mL protease and subsequently with 1 mg/mL collagenase-II at 37 °C for 4 h. The isolated HPCs were grown at 37 °C and 5% CO_2_ to 80% confluence in DMEM High Glucose supplemented with 1% L-glutamine, 1% penicillin/streptomycin, 50 μg/mL gentamicin and 10% Fetal Bovine Serum (FBS).

### 2.5. Cell Treatment

Cells were left untreated (CTL) or stimulated with TNF-α, or treated, for the required time, with different concentrations of GlcNAc and GlcNAc NP and stimulated with TNF-α. Experiments were independently repeated at least three times. The concentration of GlcNAc NP was calculated taking into consideration only the molecule without considering PVP, which can be negligible as the GlcNAc: PVP ratio was 20:1.

### 2.6. MTS Assay

To assess GlcNAc and GlcNAc NP effect on HPCs cell viability at different concentrations and time points, an MTS (3-(4,5-dimethylthiazol-2-yl)-5-(3-carboxymethoxyphenyl)-2-(4-sulfophenyl)-2H-tetrazolium)-based colourimetric assay was performed. Briefly, 8 × 10^3^ cells per well were seeded in a 96-well plate. To align cell cycle progression, the day after seeding, cells were starved overnight in a reduced serum medium. Cells were then left untreated (CTL) or treated with GlcNAc and GlcNAc NP for 24, 48, and 72 h. After each time point, 100 µL MTS solution was added to the wells. Spectrophotometric absorbance was directly measured at 492 nm after 3 h incubation.

### 2.7. RNA Extraction and Reverse-Transcription

Total RNA was extracted from untreated and treated HPCs, with Blood/Tissues Total RNA extraction kit, and reverse-transcripted according to the manufacturer’s instructions using Improm II enzyme.

### 2.8. Quantitative Real-Time PCR

Quantitative Real-Time Polymerase Chain Reaction (RT-PCR) analysis was carried out using an ABI Prism 7300 (Applied Biosystems, Thermo Fisher Scientific, Waltham, MA, USA). Amplification was performed using the SensimixPlus SYBR Master mix. Primers (Table 1) were synthesised by Bio-Fab research and designed using Primer Express software v1.4.0 (Applied Biosystems). Data were analysed by a 2^−ΔΔCt^ method based on the determination of the transcript abundance relative to the 18S housekeeping gene [28].

### 2.9. Immunofluorescence Analysis

Collagen-II protein was visualized by immunofluorescence. HPCs were plated at a density of 8 × 10^3^/cm^2^, and left untreated (CTL) or treated, for 72 h with different concentrations of GlcNAc and GlcNAc NP. Cells were washed in PBS, fixed in 4% paraformaldehyde in PBS for 15 min at 4 °C, and permeabilized with 0.5% Triton-X 100 in PBS for 10 min at room temperature (RT). Then, cellular proteins were blocked with 3% bovine serum albumin in PBS for 30 min, and cells were incubated for 1 h with anti-Collagen-II (1:100) primary antibody. Cells were washed with PBS and then incubated with Alexa Fluor 595 donkey anti-mouse red secondary antibody (1:400) for 1 h. Cells were washed and then stained with DAPI to visualise the nuclei. All these steps were performed at RT. The images were captured by a Leica DM IL LED optical microscope, using an AF6000 modular microscope (Leica Microsystem, Milan, Italy).

### 2.10. ELISA

The IL-6 and IL-8 amount in the treated and untreated HPC supernatant was determined using ELISA kits according to the manufacturer’s instructions. Optical Density (OD) absorbance was measured by a microplate reader (NeBiotech, Holden, MA, USA) at 450 nm.

### 2.11. Densitometric Analysis

The free software ImageJ v1.52t (https://imagej.nih.gov/ij/, accessed on 1 July 2020) was used to perform densitometric analysis of protein Collagen-II expression. For each cell culture condition, the integrated fluorescence density values obtained in immunofluorescence experiments were considered.

### 2.12. Statistical Analysis

All data were obtained from at least three independent experiments, performing each experiment either in duplicate or in triplicate. Data were statistically analysed using Prism 5.0 software (GraphPad Software, San Diego, CA, USA), with two-way repeated measures analysis of variance (ANOVA) with Bonferroni’s multiple comparison tests and reported. *p*-value < 0.05 was considered significant.

## 3. Results and Discussion

### 3.1. Physical Characterization of GlcNAc Nanoparticles

The synthesis of GlcNAc NP required the use of a wet mixture of 20 g of GlcNAc and PVP carrying out a planetary ball miller at room temperature (25 °C). After the process, the wet mixture was dried under a vacuum oven, yielding about 18 g of dry powder that was stored at room temperature.

The DLS analysis by the intensity of scattering, performed on samples from three independent preparations, gave three overlaying curves, showing that the distribution of GlcNAc NPs is monomodal and the Z-Average resulted in (201.6 ± 5.6) nm with a Polydispersity index (PI) of 0.378 ± 0.0187 (Figure 1). The analysis of the autocorrelation functions of the three samples showed that the curves were overlayed and the intercepts with the *y* axis are between 0.7 and 0.9. This result suggests that samples were analysed at the appropriate concentration and the multiple scattering effect was neglectable (Figure 2). Moreover, to receive information about the relative percentage of the population of nanoparticles with different sizes, the samples were analysed by MALDS. This analysis revealed three different populations of about 12 nm, 111 nm, and 410 nm, representing 5.3%, 46%, and 48.7%, respectively, of the total mass. However, the Z-average estimated with a cumulant method by fitting the distribution and autocorrelation confirmed the weighted average of the nanoparticles’ size is 201.6 nm. These data suggested that the GlcNAc NPs represent 96% of the total mass of the synthesized nanomaterial and it can be considered pure organic material. Therefore, the biological effects observed on the cells, below reported, can be ascribed to the glucosamine nanoparticles, considering that the surfactant is negligible, given the low percentage.

### 3.2. Biological Activity of GlcNAc Nanoparticles

#### 3.2.1. The Effects of GlcNAc NP on Chondrocyte Viability

To ascertain the safety of the GlcNAc NP on human primary chondrocytes, cells were treated with different concentrations of nanoparticles and analysed by MTS assay, comparing the cell viability of GlcNAc NP with both GlcNAc and untreated cells. As shown in Figure 3, GlcNAc NP was never detrimental neither at concentration nor time analysed (Figure 3). These findings suggest that the ball milling process has no negative effect on nanoparticle preparation; consequently, the NPs do not have detrimental effects on cells.

#### 3.2.2. The Effects of GlcNAc NP on the Modulation of Pro-Inflammatory Genes

OA, previously considered a non-inflammatory disease, is now defined as low-grade inflammation pathology [29]. In the synovial fluid of OA patients, several pro-inflammatory mediators such as cytokines and chemokines have been found [30]. In particular, the cytokine IL-6 and the chemokine IL-8 are present in a very large amount compared with that present in the synovial fluid of health subjects and at the same high concentration detected in rheumatoid arthritis synovial fluid [30]. IL-6 and IL-8 are among the first mediators produced by synoviocytes and are released in the synovial fluid [31], where they stimulate chondrocytes to auto-produce these and other pro-inflammatory molecules leading to cartilage degeneration.

To evaluate the effectiveness of GlcNAc NP in counteracting the inflammation, human primary chondrocytes were pretreated with several concentrations of both GlcNAc and GlcNAc NP and then stimulated with TNF-α to simulate the inflamed synovial environment. TNF-α stimulation increased the expression level of mRNA coding for IL-6 and IL-8, GlcNAc was able to decrease the IL-6 production both at 1 mM and 0.5 mM concentrations, whereas was effective in down-regulate IL-8 mRNA only at 1 mM concentration (Figure 4). GlcNAc NP was effective to decrease IL-6 and IL-8 mRNA levels up to 0.5 μM concentration (Figure 4).

The same experiment was conducted to analyse the production of IL-6 and IL-8 at the protein level. The supernatant of chondrocytes cell culture stimulated with TNF-α and treated with only the two lower concentrations of GlcNAc and GlcNAc NP (1 μM and 0.5 μM) was used to analyse the protein synthesis of IL-6 and IL-8. GlcNAc NP decreased the IL-6 production below the CTL level at both analysed concentrations, whereas GlcNAc was not able to decrease the IL-6 production, confirming the result obtained in mRNA modulation analysis (Figure 5). The treatment with 1 μM GlcNAc, which was ineffective to modulate gene expression level, was able to decrease the production of IL-8 at the CTL level, whereas 0.5 μM was still effective even if to a lesser extent. The effects of GlcNAc NP on IL-8 production confirmed the results obtained in gene expression modulation experiments, the nanoparticles were effective to decrease the IL-8 production both at 1 μM and 0.5 μM concentrations (Figure 5). However, the 0.5 μM GlcNAc NP was still much more effective than 0.5 μM GlcNAc in decreasing IL-8 secretion. The presence of AU-reach elements (ARE) in the 3′-Untranslated Region (UTR) of pro-inflammatory cytokines has been shown in several types of cells [32,33]. These elements are involved in the stabilization of interleukin mRNAs; in particular, the stabilization of IL-8 mRNA has been shown in differentiated monocytes [32]. Moreover, the stabilization of pro-inflammatory cytokine mRNAs has been found in presence of some stimuli such as stress and inflammation [34]. In chondrocytes, the binding of molecules at 3′-UTR of both anabolic and catabolic mRNAs has been shown to be effective in both the stabilization and destabilization of messengers [35]; thus, we suggest that GlcNAc can have a destabilizing effect on IL-8 mRNA, leading to a strong decrease in the IL-8 protein expression, which does not correspond to the inhibition of transcription, considering that GlcNAc at mRNA level was effective at 1 mM concentration and not at 1 μM and 0.5 μM. On the other hand, the nanoparticles were effective at all analysed concentrations.

#### 3.2.3. The Chondroprotective Effects of the GlcNAc NP

The cartilage extracellular matrix (ECM) is avascular, aneural, and populated by a single cell type, the chondrocyte, which does not exhibit mitotic activity and is responsible for the maintenance of cartilage homeostasis through a low turnover of ECM components [36]. The ability to perform this low turnover repair declines with OA, leading to progressive loss of several molecules such as collagen type II. For this reason, a potential “disease-modifying” molecule must show the ability to stimulate anabolic activity in chondrocytes as well as to decrease catabolic activity.

To verify whether GlcNAc and GlcNAc NP were able to stimulate collagen type II, cells were treated with both forms of GlcNAc, and then the collagen type II was evidenced by immunofluorescence and compared to the production in untreated cells. The findings showed that the GlcNAc was not able to increase collagen type II production, whereas, even if not to a statistically significant level, GlcNAc NP was able to stimulate collagen production (Figure 6).

Previously, GlcN and mainly GlcNAc administrations have been shown to have chondroprotective activity by inhibiting the collagen type degradation [5,22], whereas GlcN has rarely been demonstrated to stimulate collagen type II synthesis when administered at high dosage [37]. For this reason, we evaluated the ability of GlcNAc NP to inhibit metalloproteases (MMPs) production. Human primary chondrocytes were treated with 1 μM and 0.5 μM of both GlcNAc and GlcNAc NP finding that, at these concentrations, GlcNAc was not able to decrease the collagenase MMP-1 stimulated by TNF-α, whereas GlcNAc NP reduced the MMP-1 mRNA production at CTL level (Figure 7). Moreover, the GlcNAc NP was able to decrease also the stromelysin MMP-3, which is involved in the degradation of non-collagenous components of ECM, whereas GlcNAc, at low concentration, was not able to decrease the MMP-3 mRNA levels (Figure 7).

## 4. Conclusions

Different types of organic nanoparticles of pharmaceutical interest have been obtained on the scale of milligrams, by the planetary ball milling process. Mostly, these nanomaterials are composed of drugs loaded on liposomes or polymers, such as chitosan, or nanosponge [38,39,40,41,42]. In the present manuscript, different outcomes have been reported. Firstly, we obtained, on a scale of grams, stable and pure GlcNAc nanoparticles with an average size of about 200 nm with a PI value of 0.38. Furthermore, DLS analyses of the sample stored at room temperature, repeated after six mounts, did not show maturation or aggregation processes. The availability of a high amount of nanoparticles is important for preclinic studies and to show that process of production is scalable.

Secondly, the analyses of the biological activity of GlcNAc NP showed that the size of the aminosugar improves the effect to counteract the pathways involved in the onset and progression of osteoarthritis. In particular, GlcNAc NP decreases the production of pro-inflammatory modulators IL-6 and IL-8 at a concentration of 2000 folds lower compared to GlcNAc microparticles. This is a very interesting result, meaning that the GlcNAc in nanoform could be administered at a lower concentration than those currently used. Moreover, the finding that the effectiveness of GlcNAc to decrease the IL-8 protein production at concentrations that were ineffective in modulating the IL-8 mRNA expression levels is intriguing and suggests that the GlcNAc could destabilize the IL-8 mRNA, inhibiting its translation and thus decreasing IL-8 secretion.

Finally, in the present manuscript we found that GlcNAc in nanoform is able to stimulate collagen type II production, whereas, previously, GlcNAc has never shown this effectiveness. Moreover, GlcNAc NP also shows a chondroprotective activity due to the inhibition of metalloproteases, MMP-1 and MMP-3, once again at a concentration much lower than that resulted effective in other studies. In conclusion, the GlcNAc in nanoform is very effective both as an anti-inflammatory and chondroprotective agent. These results suggest that GlcNAc NP is a promising candidate for the prolonged treatment of osteoarthritis. However, preclinical studies in animals will be necessary to confirm the effects in a chronic model of pathology.

## Figures and Tables

**Figure 1 bioengineering-10-00343-f001:**
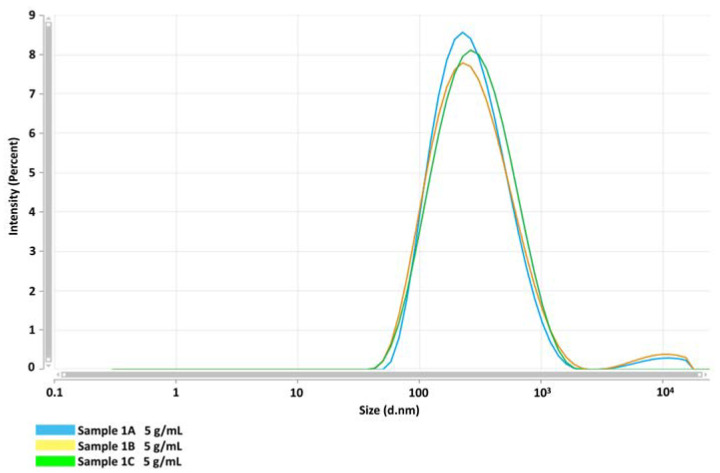
Size distribution of GlcNAc NP suspension at 5 g/mL by DLS at a fixed angle of scattering, expressed as Z-average (d.nm). Three independent preparations were carried out to analyse the size and distribution of NPs. Sample 1A (blue line), Sample 1B (yellow line), and Sample 1C (green line).

**Figure 2 bioengineering-10-00343-f002:**
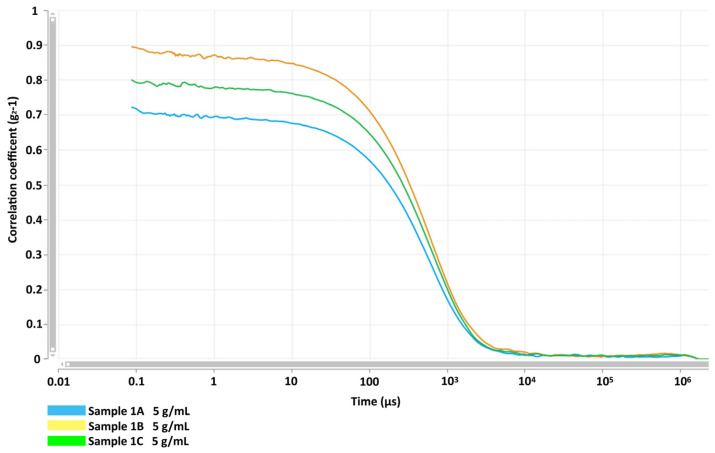
GlcNAc NP autocorrelation functions regarding the effects of sample dilutions and scattering at a fixed angle. Samples were obtained from three independent preparations: Sample 1A (blue line), Sample 1B (yellow line), and Sample 1C (green line). The intercepts of curves show that all coefficients are inside the range from 0.7 to 0.9 and the curve slopes suggest a low effect of scattering among particles with different sizes.

**Figure 3 bioengineering-10-00343-f003:**
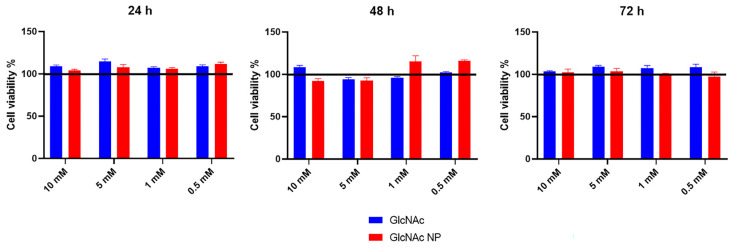
The MTS colourimetric method was performed to assess the cell viability of HPCs treated with four different concentrations of GlcNAc and GlcNAc NP (10 mM, 5 mM, 1 mM, and 0.5 mM), for 24, 48, and 72 h. Cell viability of treated samples was normalized to the untreated cells, which is reported as 100% and represented in the graphs by a black horizontal line.

**Figure 4 bioengineering-10-00343-f004:**
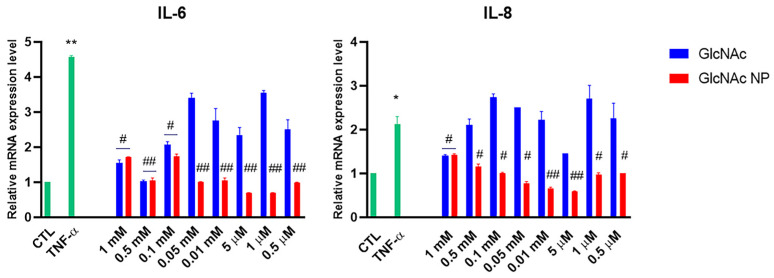
Effects of GlcNAc and GlcNAc NP on interleukins mRNA expression levels under TNF-α stimulus. Cells were left untreated (CTL), stimulated with 10 ng/mL TNF-α for 30 min, or treated with different concentrations of GlcNAc and GlcNAc NP for 1 h and then stimulated with 10 ng/mL TNF-α for 30 min. After treatments, mRNA was extracted and analysed by RT-PCR. IL-6 and IL-8 mRNA levels were reported as relative mRNA expression levels with respect to 18S mRNA (2^−ΔΔCt^ method). Results are expressed as mean ± standard deviation (SD) of data obtained by three independent experiments. * *p* < 0.05; ** *p* < 0.01 vs. CTL; # *p* < 0.05; ## *p* < 0.01 vs. TNF-α.

**Figure 5 bioengineering-10-00343-f005:**
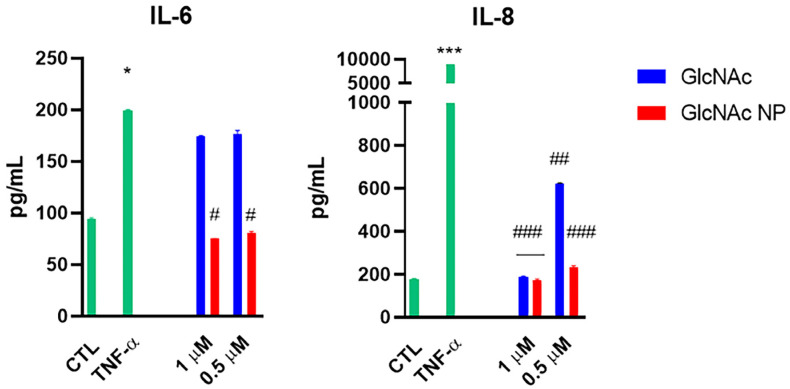
Effects of GlcNAc and GlcNAc NP on interleukins secretion in the culture medium under TNF-α stimulus. Cells were left untreated (CTL), stimulated with 10 ng/mL TNF-α for 30 min, or treated with two concentrations of GlcNAc and GlcNAc NP for 1 h and then stimulated with 10 ng/mL TNF-α for 1 h. After treatments, cell supernatants were collected and analysed by ELISA. The results are reported as pg/mL and are expressed as mean ± SD of data obtained by three independent experiments. * *p* < 0.05 and *** *p* < 0.005 vs. CTL; # *p* < 0.05, ## *p* < 0.01 and ### *p* < 0.005 vs. TNF-α.

**Figure 6 bioengineering-10-00343-f006:**
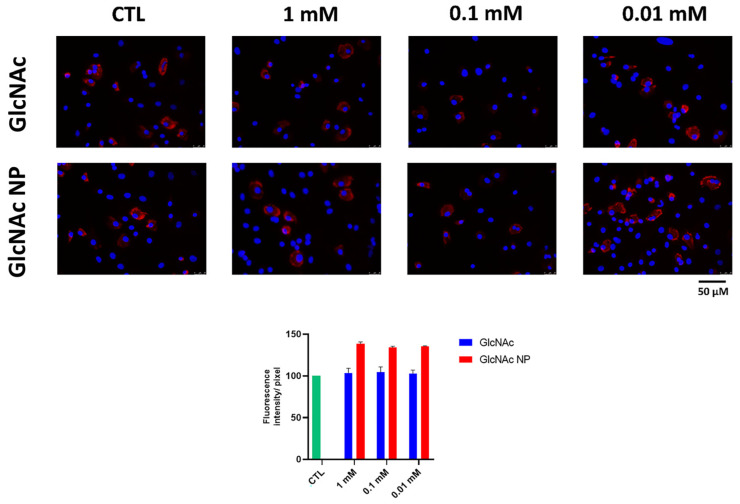
Effects GlcNAc and GlcNAc NP on Collagen-II production. Cells were treated with 1 mM, 0.1 mM, and 0.01 mM of GlcNAc and GlcNAc NP, respectively, for 72 h and then analysed by immunofluorescence using an anti-Coll-II primary antibody and Alexa Fluor 594 (red) secondary antibody. Nuclei were stained with DAPI (original magnification 40×). The histogram represents the pixel intensities in the region of interest, obtained by ImageJ.

**Figure 7 bioengineering-10-00343-f007:**
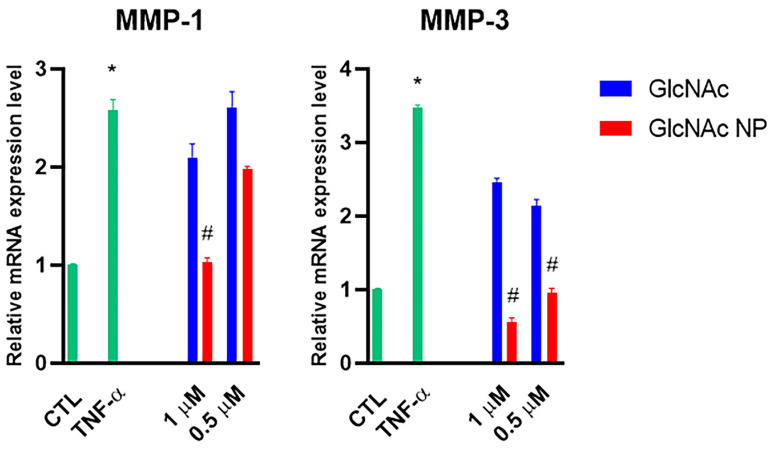
Effects of GlcNAc and GlcNAc NP on MMPs under TNF-α stimulus. Cells were left untreated (CTL), stimulated with 10 ng/mL TNF-α for 24 h, or stimulated with 10 ng/mL TNF-α and treated with 1 μM and 0.5 μM of GlcNAc and GlcNAc NP for 24 h. After 24 h treatments, mRNA was extracted and analysed by RT-PCR. MMP-1 and MMP-3 mRNA levels were reported as relative mRNA expression levels with respect to 18S mRNA (2^−ΔΔCt^ method). Results are expressed as mean ± SD of data obtained by three independent experiments. * *p* < 0.05 vs. CTL; # *p* < 0.05 vs. TNF-α.

**Table 1 bioengineering-10-00343-t001:** RT-PCR primer sequences.

Gene Accession Number	Primer Forward Primer Reverse
IL-6 NM_000600	5′-GATGGATGCTTCCAATCTG-3′ 5′-CTCTAGGTATACCTCAAACTCC-3′
IL-8 NM_000584	5′-GACATCAAAGAAGGACTTG-3′ 5′- GCCACAATTTCAGATCCTG-3′
MMP-1 NM_002421	5′-GATGGACCTGGAGGAAATCTTG-3′ 5′-TGAGCATCCCCTCCAATACC-3′
MMP-3 NM_002422	5′-CCTGGTACCCACGGAACCT-3′ 5′-AGGACAAAGCAGGATCACAGTTG-3′
18S NM_003286	5′-CGCCGCTAGAGGTGAAATTC-3′ 5′-CATTCTTGGCAAATGCTTTCG-3′

## Data Availability

Not applicable.

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
