# Peer review of "The Formulation of the N-Acetylglucosamine as Nanoparticles Increases Its Anti-Inflammatory Activities: An In Vitro Study"

_bioengineering, 2023, doi:10.3390/bioengineering10030343_

Round 1
Reviewer 1 Report
Please find the attachment for my comments

Author Response
REVIEWER 1
The manuscript entitled "The formulation of the N-acetylglucosamine as nanoparticles increases its anti-inflammatory activities: an in vitro study" deals with evaluation of the effects of N-Acetyl glucosamine in nanoform (GlcNAc NP) in an in vitro model of osteoarthritis. It is well written and organized, however it needs some revision before it can be considered for publication:
- I would like to recommend to include a separate section for materials and providing a list of chemicals used.
Thank you for your comment, a Materials section has been added to the revised version of the manuscript.
- Did authors have performed any blend flow properties before or after ball milling? If not justify.
Experiments have not been carried out to evaluate the blend flow properties. The aim of this work is to study the anti-inflammatory effects of the N-acetylglucosamine NP in in vitro model, in order to compare the nano- to native- form. Moreover, we were aimed to find a method to obtain nanoparticles without the use of other auxiliary agents, such as lipids, polymers etc. We used only a biocompatible dispersant PVP, in very low concentration (GlcNAc:PVP =20:1). In future, the blend flow property analysis will be mandatory to perform preclinical study on animal model.
- I would suggest to include SEM or TEM analysis for particle size.
Physical and morphological characterization of GlcNAc and GlcNAc NP by SEM have been done. This analysis showed that GlcNAc in bulk form had three principal morphologies: agglomerates with sizes up to 2.5 µm, particles with irregular shape with an average diameter up to 850 nm and elongated shape particles with an average diameter up to 300 nm. While, GlcNAc-NPs had some shapes: spherical, elongated and irregular one. We apologize, but we cannot show these results because they are reported in another manuscript that is going for publication in International Journal of Molecular Sciences (ijms-2201229).
- Include some of the relevant recent references such as 10.1016/j.carbpol.2021.118858, 10.1016/j.arabjc.2023.104551, 10.1021/acschemneuro.1c00022, 10.1016/j.ijbiomac.2023.123458 etc
Thank you very much for your suggestion, we added some of the above references. Two of them were not added because they describe inorganic NP (Au), while our manuscript regards organic ones.
- In general, it is well written, but there are some grammars and typing/spacing errors, a few of them are at line no 105 include the respective citation not DOI number, these should be corrected. The English is generally satisfactory but a native speaker should read the paper and correct some sentences
A native speaker read and corrected the revised manuscript.
- The outcomes of this study must be systematically described in conclusion section.
The Conclusions section has been reformulated following your suggestion.
This work is interesting with novel objective and the authors have done a number of experiments to establish their hypotheses. Still the article has some queries needs to be addressed before publishing.
We would like to thank the reviewer for his accurate revision of our manuscript hoping that it is now suitable for publication.

Reviewer 2 Report
This paper is well planned and constructed I suggest it for publication on Bioengineering
Author Response
This paper is well planned and constructed I suggest it for publication on Bioengineering.
Thank you very much for your comment.
Reviewer 3 Report
The manuscript about “The formulation of the N-acetylglucosamine as nanoparticles increases its anti-inflammatory activities: an in vitro study”.
The GlcNAc in nano-form showed better performance than GlcNAc, at concentrations lower than those reached in the joints after oral administration to patients of 1.5 g/die of glucosamine.
Findings show that SCP-Car-NCs treatment reduced the viability and increased apoptosis in the U266 cells, providing a new insight on SCP-Car-NCs' potential for usage in the future to treat multiple myeloma.
Specific comments need to be addressed
1. The abstract not provided the sufficient details so it has to be revised based on the findings mainly about the inflammation.
2. The results part must concise based on the findings.
3. Hypothesis, objective and problem statement need to be provided based on the methods.
4. Figure description is not satisfactory hence I would suggest authors to provide adequate details in the figure legends.
5. The results shows only cytokines IL-6 and IL-8 but other key cytokines were are not studied so need to provide the justification of IL-6 and IL-8?
6. The quality of all the figures must be improved since some of the not clear.
7. In the results, The HPCs were treated with four different concentrations of GlcNAc and GlcNAc NP (10 mM, 5 mM, 1 mM, 0.5 mM), for 24, 48, and 72 h, so authors should explain how they calculated the nanosamples in mM?.
Author Response
REVIEWER 3
The manuscript about “The formulation of the N-acetylglucosamine as nanoparticles increases its anti-inflammatory activities: an in vitro study”.
The GlcNAc in nano-form showed better performance than GlcNAc, at concentrations lower than those reached in the joints after oral administration to patients of 1.5 g/die of glucosamine.
Findings show that SCP-Car-NCs treatment reduced the viability and increased apoptosis in the U266 cells, providing a new insight on SCP-Car-NCs' potential for usage in the future to treat multiple myeloma.
Specific comments need to be addressed
- The abstract not provided the sufficient details so it has to be revised based on the findings mainly about the inflammation.
Thank you for your suggestion, we amended the Abstract, reporting more information on the anti-inflammatory effect of the NPs.
- The results part must concise based on the findings.
Thank you for your suggestion, the Results section include also Discussion, for this reason is not concise, in order to give the readers the necessary explanations.
- Hypothesis, objective and problem statement need to be provided based on the methods.
We rephrased the final part of the Introduction section, describing more specifically aim of the present paper.
- Figure description is not satisfactory hence I would suggest authors to provide adequate details in the figure legends.
The figure legends have been implemented.
- The results shows only cytokines IL-6 and IL-8 but other key cytokines were are not studied so need to provide the justification of IL-6 and IL-8?
In synovial fluid of OA patients, there are several pro-inflammatory mediators, anyway the most expressed are exactly IL-6 and IL-8 (1). For this reason, to evaluate the effectiveness of an anti-inflammatory molecule, these are the first that must be analyzed (2,3). In the present manuscript we would find the lowest concentration of GlcNAc NP showing an anti-inflammatory activity, and we obtained this result. In future, further experiments to analyze in more dept the anti-inflammatory activity will be carried out.
- Kaneko S, Satoh T, Chiba J, Ju C, Inoue K, Kagawa J. Interleukin-6 and interleukin-8 levels in serum and synovial fluid of patients with osteoarthritis. Cytokines Cell Mol Ther. 2000 Jun;6(2):71-9. doi: 10.1080/13684730050515796. PMID: 11108572.
- Park K, Lee JH, Cho HC, Cho SY, Cho JW. Down-regulation of IL-6, IL-8, TNF-α and IL-1β by glucosamine in HaCaT cells, but not in the presence of TNF-α. Oncol Lett. 2010 Mar;1(2):289-292. doi: 10.3892/ol_00000051. Epub 2010 Mar 1. PMID: 22966296; PMCID: PMC3436374.
- Lopreiato M, Cocchiola R, Falcucci S, Leopizzi M, Cardone M, Di Maio V, Brocco U, D'Orazi V, Calvieri S, Scandurra R, De Marco F, Scotto d'Abusco A. The Glucosamine-derivative NAPA Suppresses MAPK Activation and Restores Collagen Deposition in Human Diploid Fibroblasts Challenged with Environmental Levels of UVB. Photochem Photobiol. 2020 Jan;96(1):74-82. doi: 10.1111/php.13185. Epub 2019 Dec 17. PMID: 31769510.
- The quality of all the figures must be improved since some of the not clear.
All the figures have been substituted with others having an increased resolution (600 px/cm)
- In the results, The HPCs were treated with four different concentrations of GlcNAc and GlcNAc NP (10 mM, 5 mM, 1 mM, 0.5 mM), for 24, 48, and 72 h, so authors should explain how they calculated the nanosamples in mM?
To prepare nanoparticles, the amount of dispersant PVP respect the N-acetylglucosamine was in a ratio GlcNAc: PVP = 20:1, which can be negligible. We calculated the mM taking in consideration only GlcNAc without considering PVP. In this way, the effective concentration of GlcNAc is lower than mM reported, indicating that the NPs are more effective that 5 mM indicated in the manuscript. We added this description in Mat & Met section, 2.5 paragraph.
We would like to thank the reviewer for his exhaustive comments to our manuscript hoping that it is now suitable for the readers of the journal.

Round 2
Reviewer 1 Report
The authors have addressed all of the comments raised, and the manuscript can be accepted in its current form.
Reviewer 3 Report
The authors have addressed all the queries nicely therefore the revised version of manuscript is acceptable for publication.